# Analysis of Nursing Education Curricula in Spain: Integration of Genetic and Genomic Concepts

Estel·la Ramírez-Baraldes [1,2,3,4], Katerin Ariza-Martin [1], Daniel García-Gutiérrez [1,2,3,*] and Cristina García-Salido [1,2,3]

1    Departamento de Enfermería, Facultad de Ciencias de la Salud de Manresa, Universitat de Vic—Universitat Central de Catalunya (UVic-UCC), Av. Universitària, 4-6, 08242 Manresa, Barcelona, Spain; eramirez@umanresa.cat (E.R.-B.); katerinam@correu.umanresa.cat (K.A.-M.); cgarcia@umanresa.cat (C.G.-S.)

2    Grupo de Investigación en Simulación e Innovación Transformativa (GRIST), UVic-UCC, Av. Universitària, 4-6, 08242 Manresa, Barcelona, Spain

3    Instituto de Investigación e Innovación en Ciencias de la Vida y de la Salud de la Cataluña Central (Iris-CC), 08500 Vic, Barcelona, Spain

4    Unitat de Cures Intensives, Althaia Xarxa Assistencial Universitària de Manresa, 08243 Manresa, Barcelona, Spain

*    Correspondence: dgarcia04@umanresa.cat

**Abstract:** The integration of genetics and genomics into nursing practice is essential for addressing genetic pathologies and providing personalized patient care. This study aims to analyze the nursing education curricula across Spanish universities to understand how genetic and genomic concepts are incorporated. Using the "Qué estudiar y dónde en la Universidad" (QEDU) database, an official source provided by the Spanish Ministry of Universities, a systematic review of 4720 teaching guides from 118 university centers was conducted. Our findings reveal that 12 centers do not include any genetic or genomic content, while 43% of the remaining centers focus solely on basic genetics. In contrast, 57% cover advanced topics, such as genetic counseling and pharmacogenetics. Teaching methodologies predominantly involve lectures and seminars, with limited practical training. On average, three to six ECTS credits out of the total 240 required for the Nursing Degree are dedicated to genetics and genomics. This study highlights the need for curriculum updates and standardization to ensure comprehensive training in these crucial areas. Improved integration of genetic and genomic education will better prepare nursing professionals to meet the evolving demands of modern healthcare. This study was not registered.

**Keywords:** genetics; genomics; genetic counseling; nursing education; nurse

## 1. Introduction

The integration of genetics and genomics into clinical nursing practice represents a pivotal advancement toward personalized healthcare, enabling tailored treatments and preventive strategies based on an individual's genetic profile. As frontline healthcare providers, nurses must acquire comprehensive knowledge and skills in these areas to deliver effective patient care, including genetic counseling and education [1,2].

Genetics is the branch of biology that focuses on the mechanisms governing the transmission and expression of hereditary traits in living organisms. Genomics, on the other hand, encompasses the study of all genes within the human genome and investigates the structure, function, evolution, and mapping of genomes [2,3]. Recent breakthroughs in genomic research and technology have significantly enhanced our understanding of the etiology and progression of genetic disorders. Translating these scientific advancements into clinical practice is essential for providing personalized care that includes targeted therapies and preventive measures tailored to each individual's genetic makeup [4].

Genetic counseling is a critical aspect of healthcare, guiding individuals and families in understanding genetic risks, making informed decisions about testing, and interpreting genetic information for health and family planning [1]. In countries where genetic counseling is a recognized profession, specialists extensively trained in genetics, psychosocial counseling, and ethics typically perform this role. Nurses also contribute by providing foundational genetic education, initial risk assessment, and emotional support within their broader patient care responsibilities, particularly in specialties like oncology and cardiology. Unlike certified genetic counselors, nurses integrate genetic counseling into general healthcare duties, focusing on practical aspects such as preparing patients for genetic testing, explaining results, and guiding follow-up care. However, in many countries, including Spain, regulatory frameworks limit this role, as genetic counseling by nurses lacks formal recognition and structure [5,6].

In Spain, genetic services are primarily available in specialized centers and tertiary hospitals, administered by professionals with specialized genetics training. The integration of genetic testing and counseling into primary care is limited, restricting patient access outside specialized settings [6–8]. Consequently, the role of nurses in genetic counseling is still emerging and lacks a clear regulatory framework to support their active involvement.

Contrastingly, in countries like the United Kingdom and the United States, nurses with genetics training have established roles in genetic counseling and testing. Spain lacks such regulatory support for nursing roles in this domain, and without a formally recognized genetic counseling profession, responsibilities typically fall to physicians with genetic expertise rather than dedicated counselors [7,8]. This gap limits nurses' potential contributions to genetic and genomic care, as their involvement remains limited and regionally variable. While some specialized nurses may incorporate genetic counseling into their practice, this is less common than in countries with more established nursing roles in genetics [9,10].

The application of genetic techniques in nursing facilitates accurate disease diagnosis and the identification of individuals who are carriers or are at risk of developing genetic conditions. This knowledge is pivotal for nursing practice, particularly in the realms of personalized care, disease prevention, and management. Genetic counseling provided by nurses assists individuals and families in comprehending genetic information, assessing risks, making informed decisions about genetic testing, and receiving emotional support during challenging times [10,11].

For instance, nurses specializing in oncology and cardiology are already integrating genetic counseling into their practice by explaining genetic testing processes, obtaining informed consents, addressing patient queries, collecting genetic samples, and effectively communicating test results. This specialized role requires nurses to possess not only theoretical knowledge, but also practical skills and ethical awareness to support patients in understanding the implications of their genetic information and making informed health decisions [12].

International initiatives, such as the Human Genome Project (1990–2003), have been instrumental in mapping and sequencing the human genome, thereby accelerating the identification of disease-associated genes and fostering biomedical research through collaborative data sharing. In 2017, the Global Nursing Alliance (G2NA) was established to support the integration of genomics into nursing practice, promoting international collaboration and leadership in this field [13]. Similarly, the GenoNurse project in Europe aims to create educational models and define competencies for nursing professionals based on best practices and expert consensus [14].

In Spain, the absence of a recognized genetic counseling profession has impacted the ability of nurses to engage formally in genomic healthcare roles. Nurses can fill these gaps if provided with appropriate training and regulatory support, as demonstrated by the experience of other countries. The Spanish healthcare system has seen efforts to address the evolving demands of healthcare, including genetics and genomics, with widespread strategies like the Spanish Science, Technology, and Innovation Strategy, which

aims to strengthen the healthcare system and foster innovation across multiple fields [15]. However, these strategies do not explicitly address the integration of nursing roles in genetic counseling, highlighting the need for specific regulatory and educational frameworks in this area.

The anticipated transformation of healthcare services through genetics and genomics offers nurses opportunities for leadership and positive impacts on the healthcare system. Achieving this transformation necessitates sustained investment in nursing education, infrastructure, and policy development [16–18]. Recognizing the imperative for standardized genetic competencies, organizations like EuroGentest and the European Society of Human Genetics (ESHG) developed basic competencies in 2010 to guide European countries in adapting their educational frameworks accordingly [1,2].

Despite these advancements, literature reviews indicate that nurses are generally inadequately prepared to apply genetic and genomic knowledge in clinical settings, underscoring a significant gap in genomic competencies [3,4,7–9,16,17]. Studies have revealed that the nursing faculty often possesses genomic knowledge comparable to that of students, highlighting deficiencies in educational preparedness [10–12]. Recent research emphasizes the urgent need for effective educational programs and curricular reforms to enhance nurses' readiness concerning genetics and genomics [13,14,18–23].

Study [23] evaluated the genomic content in European nursing curricula and found low levels of genomic literacy among nursing students, attributed to knowledge deficits and insufficient curricular integration. The study identified barriers, including inadequate genomic knowledge among faculty members, limited recognition of the importance of genomics, overcrowded curricula, and the absence of regulatory requirements for genomic competency. These findings suggest that genetics and genomics are not yet fully integrated into nursing education across various countries, mirroring the challenges identified in previous years.

In the context of Spain, significant variability exists in the content and structure of nursing curricula concerning genetics and genomics [5,6]. However, there is a lack of specific studies or reviews that assess the extent of genetic and genomic integration in Spanish nursing programs. This gap underscores the necessity for a systematic evaluation of current educational practices to ensure that nursing graduates are adequately equipped to meet the demands of modern healthcare. Addressing this need may involve integrating genetics into existing curricula or establishing specialized roles, such as the "Expert Nurse in Genetics and Genomics", to lead genomic initiatives within healthcare settings.

The establishment of an "Expert Nurse in Genetics and Genomics" role would provide a specialized resource within healthcare institutions, capable of guiding and supporting both nursing staff and students in understanding and applying genetic and genomic information in clinical settings. This role could bridge the gap between theoretical education and clinical practice by leading educational initiatives, offering mentorship, and developing resources and workshops for nursing graduates and practicing nurses. By creating a structured pathway for ongoing genetic and genomic education within healthcare facilities, the expert nurse would help ensure that both new graduates and current practitioners develop the competencies needed to integrate personalized genomic care into their practice, ultimately enhancing patient care outcomes.

To improve the overall integration of genetics and genomics in nursing education, this study aims to review the teaching guides of all Spanish universities offering a Bachelor of Nursing Degree to determine the extent of genetic and genomic content, identify institutions that provide specific training in these areas, and examine the allocated time and teaching methodologies used. This comprehensive analysis will inform recommendations for curricular enhancements to promote standardized, high-quality genetics and genomics education for nursing professionals in Spain.

Enhancing genetics and genomics education within nursing curricula is crucial for preparing nurses to contribute effectively to personalized medicine, improve patient outcomes, and advance public health objectives. By addressing the identified gaps and imple-

menting the recommended curricular reforms, nursing education in Spain can better align with the evolving demands of healthcare, ensuring that graduates possess the necessary competencies to integrate genetic and genomic information into their clinical practice. This alignment not only elevates the quality of nursing education, but also enhances patient care by enabling nurses to engage in genetic counseling, interpret genetic tests, and participate in interdisciplinary teams focused on personalized treatment strategies.

## 2. Materials and Methods

### 2.1. Study Design

A cross-sectional descriptive study was conducted through a systematic review of the teaching guides for Bachelor of Nursing programs at all Spanish universities offering this degree. The primary aim was to analyze the inclusion and depth of genetic and genomic content, as well as the teaching methodologies employed in these programs.

### 2.2. Data Sources and Search Strategy

The identification of universities and centers was based on official and publicly accessible sources to ensure the comprehensiveness and timeliness of the information:

— Ministry of Universities of Spain (UNIVbase) [24]: The official database providing updated information on universities, academic centers, and degree programs across Spain.
— "Qué Estudiar y Dónde en la Universidad" (QEDU) [25]: An official platform from the Ministry of Universities that offers detailed information on academic programs and institutions, primarily aimed at prospective students and educational professionals.

The search was conducted in June 2023, focusing on the teaching guides corresponding to the 2023–2024 academic year. The search terms included "Nursing Degree", "genetics", "genomics", "genetic counseling", and "pharmacogenomics" to identify relevant courses and content.

### 2.3. Inclusion and Exclusion Criteria

The following criteria were applied to ensure the relevance and quality of the data collected:

— Inclusion Criteria:
  - Spanish universities offering the Nursing Degree during the 2023–2024 academic year.
  - Public, private, and affiliated centers were included to provide a comprehensive overview of the national educational landscape.
  - Teaching guides that explicitly mentioned genetics and/or genomics content, whether in compulsory or elective courses.
— Exclusion Criteria:
  - Universities or centers not offering the Nursing Degree during the study period.
  - Teaching guides that were publicly inaccessible or presented technical issues preventing their review.
  - Courses without relevant content in genetics or genomics.
  - Centers duplicated in the databases or that had discontinued offering the Nursing Degree in the 2023–2024 academic year.

### 2.4. Data Collection Procedure

A systematic protocol was followed to collect and analyze the data:

— Identification of Academic Centers: An initial list of 89 Spanish universities was compiled, of which 65 offered the Nursing Degree. Including affiliated and satellite campuses, the total number of centers rose to 118.

- Review of Teaching Guides: A total of 4720 teaching guides was downloaded and reviewed, representing the courses offered across the four years of the Nursing Degree program in the 118 identified university centers. Of these 118 centers, 12 were excluded from analysis as their teaching guides did not include any genetic or genomic content. No additional exclusions were necessary, as all other accessible data met the study's inclusion criteria.
- Selection of Relevant Courses: A predefined evaluation matrix was used consistently to identify and extract data from courses containing genetics and genomics content. This matrix guided the systematic review of each teaching guide, ensuring detailed recording of course objectives, syllabus content, teaching methodologies, and assessment methods relevant to genetics and genomics.
- Data Extraction: For each relevant course, the following information was recorded:
  - Course name.
  - Course type (compulsory or elective).
  - Year of study in which the course was offered.
  - ECTS credits allocated to the course.
  - Teaching methodologies employed.
  - Specific details on genetic and genomic content. In preparation for this study, we identified a set of key genetics and genomics competencies commonly recognized in the literature [1,2,17,18,20–22]. This predefined list informed our review process, helping us evaluate the comprehensiveness of curriculum content in meeting current standards in genetic and genomic education for nursing.

### 2.5. Data Analysis

The data were analyzed using descriptive statistics. Frequencies and percentages were calculated for categorical variables, while measures of central tendency were used for continuous variables. The analysis was performed using IBM SPSS Statistics version 26.

- Coverage of genetics and genomics: The percentage of centers including these topics in their curricula was determined.
- Depth of content: Courses were categorized based on the depth of genetic and genomic content (basic vs. advanced). In this study, "basic" content was defined as foundational genetic and genomic topics, such as introductory principles of genetics, basic gene functions, and fundamental inheritance patterns. "Advanced" content included specialized topics, like genetic counseling, pharmacogenomics, clinical applications of genomics, and methods for interpreting complex genetic information. These distinctions helped in categorizing the depth of coverage across different programs
- Teaching methodologies: The prevalence of different pedagogical approaches, such as lectures, laboratory practices, seminars, and case-based learning, was analyzed.
- Distribution of ECTS credits: The proportion of ECTS credits dedicated to genetics and genomics in relation to the total credits required for the Nursing Degree was evaluated.

### 2.6. Reliability and Validity

To enhance the reliability of the study, a review of the teaching guides was conducted independently by two researchers. Discrepancies were resolved by consensus. A pilot review of a subset of guides was conducted to ensure the suitability of the review protocol before the full-scale analysis.

## 3. Results

In this study, an exhaustive review was conducted of the teaching guides of the Nursing programs offered by Spanish universities to identify the integration of content related to genetics and genomics in the training of future health professionals. The results are presented as five main categories: coverage of genetics and genomics in the study

programs, ECTS credits dedicated to genetics and genomics, teaching methodologies, lack of standardization in curricula, and distribution of content between compulsory and optional subjects.

### 3.1. Coverage of Genetics and Genomics in the Study Programs

Of the 118 university centers that offer the Nursing Degree in Spain, it was found that 12 of them (10%) do not include genetics or genomics content in their programs. This indicates a total lack of integration in a significant proportion of educational centers. The remaining 108 centers (90%) do include some form of teaching related to these topics, although with great variations in focus and depth of content.

Within the centers that address these topics, 43% limit themselves to teaching basic genetics, with introductory content covering the fundamentals of genetics but not delving into specialized areas that are crucial for clinical practice, such as genetic counseling, pharmacogenetics, or advanced genetic diagnostic techniques. In contrast, 57% of the universities does include more advanced topics in genetics and genomics, such as genetic counseling, clinical genetics, and pharmacogenetics, although these are usually integrated into broader modules and often do not receive dedicated focus.

Of the 12 centers found to lack genetics or genomics content, these centers are distributed across a small number of universities, where some affiliates within the same university do include such topics in their programs. This highlights a discrepancy in curriculum implementation among affiliated campuses. While most universities employ curriculum committees or oversight structures intended to ensure consistency across affiliated centers, the presence of such variability indicates that these mechanisms may not fully address standardization in the inclusion of genetics and genomics. This variability in the curriculum highlights the need for a more cohesive approach within Spanish nursing programs to ensure comprehensive and consistent training across all campuses.

Our review revealed that, while basic genetics topics are commonly covered, there is considerable variability in the inclusion of specialized areas essential for clinical application. Key topics, such as genetic counseling and pharmacogenomics—identified as central based on established competencies—are frequently missing from curricula. This gap underscores a significant disparity between the current content provided in Spanish nursing programs and the competencies necessary for effectively integrating genetics and genomics into nursing practice.

### 3.2. ECTS Credits Dedicated to Genetics and Genomics

One of the most relevant findings is the scarce allocation of ECTS credits to the teaching of genetics and genomics. Of the total 240 ECTS that make up the Nursing Degree, it is estimated that content related to these areas represents between three and six ECTS in most universities that include them. This equates to approximately between 1.25% and 2.5% of the total training, which is insufficient considering the growing role of genetics and genomics in modern medicine.

This low allocation of credits underscores the need to review and expand the inclusion of these contents in the curricula. It is concerning that, in many universities, students do not receive the necessary training to adequately address advances in genomic medicine that will significantly impact healthcare in the coming years.

### 3.3. Teaching Methodologies

The analysis of the methodologies used for teaching genetics and genomics revealed that the predominant approach is theoretical, with extensive use of lectures and seminars. In many cases, it is limited to the transmission of knowledge without practical activities that allow students to develop applied competencies. This approach may hinder the development of practical skills that are essential for applying this knowledge in direct patient care.

Less than a third of the universities that teach advanced genetics content includes practical activities, such as clinical simulation, analysis of real cases, or work in specialized laboratories. These are necessary to ensure that students acquire competencies related to clinical decision making based on genetic information, which is crucial in the context of personalized medicine.

### 3.4. Lack of Standardization in Curricula

The results also highlight a lack of harmonization and standardization in the inclusion of genetics and genomics among Spanish universities. There is no common curriculum or clear national guidelines that ensure minimum training in these fields. While some universities have advanced toward the inclusion of advanced genetics topics, others limit themselves to imparting basic knowledge without a clinical focus.

This finding is particularly relevant, as the lack of a minimum training standard could generate inequalities in the competencies of future nursing professionals depending on the university where they are trained. Some students might graduate without sufficient training to apply genetic advances in clinical practice, posing a challenge to homogeneity in the quality of healthcare throughout the country.

While subjects such as Anatomy, Physiology, and Pharmacology within the Spanish nursing curricula exhibit a high degree of standardization due to national regulations and accreditation requirements established by the Ministry of Universities and professional nursing organizations [24–27], genetics and genomics lack such uniformity. The standardization in these other areas results from collaborative efforts among educational institutions, regulatory bodies, and professional associations, involving regular curriculum reviews and adherence to national educational frameworks [27–29].

Our analysis of curriculum documents reveals that explicit references to the assessment of genetic and genomic knowledge and skills are scarce. Although some universities detail assessment methods, genetics and genomics content is often evaluated indirectly as part of broader subjects without specific competency assessments. Given that assessment drives learning by signaling to students what is important [28,30,31], the absence of explicit assessment in genetics and genomics may reduce emphasis on these subjects. Incorporating targeted assessment strategies could enhance student engagement and ensure the acquisition of essential genetic and genomic competencies.

### 3.5. Distribution of Content Between Compulsory and Optional Subjects

Another important result is the distribution of content related to genetics and genomics between compulsory and optional subjects. In many universities, basic genetic concepts are integrated into core subjects such as Biology, Physiology, or Biochemistry, while more specialized content is usually offered in optional subjects. This means that not all students have access to advanced genetics training, as the optional offerings depend on their availability, students' choices, and in some cases, the university's resources.

The teaching of advanced genetics, when available, tends to be fragmented among various subjects and is not presented as a coherent unit. This makes it difficult for students to perceive the relevance of genetics as an integrated area within nursing, rather than seeing it as a marginal complement to their general training.

### 3.6. Examples of Course Content Descriptions in Genetics and Genomics

To illustrate the variability in the inclusion of genetics and genomics in nursing curricula, we present anonymized examples from different universities' syllabi:

- University A offers a compulsory course titled Fundamentals of Molecular Biology, introducing basic molecular biology principles, including DNA, RNA, proteins, inheritance, and gene expression.

- University B provides an elective course, Clinical Genetics and Genetic Counseling, delving into the clinical application of genetics, covering the detection and management of genetic diseases, the nursing role in genetic counseling, and ethical considerations.
- University C includes a compulsory course, Introduction to Genomics and Pharmacogenomics, exploring fundamental genomics concepts, the human genome, sequencing techniques, and pharmacogenomics in personalized treatments.
- University D has a compulsory course, Biology for Health Sciences, covering general biology related to human health, focusing on cellular and physiological aspects without in-depth exploration of genetics or genomics.

These examples demonstrate the diversity of how universities address genetics and genomics in their nursing programs. While some institutions offer dedicated, in-depth courses, others limit content to a basic introduction within broader subjects. This variability highlights the lack of standardization in genetics and genomics education among Spanish universities. Graduates from different institutions may possess significantly disparate competencies and levels of preparedness, impacting the equity and quality of healthcare they can provide.

## 4. Discussion

This study provides a comprehensive analysis of the inclusion of genetics and genomics content within Bachelor of Nursing curricula across Spanish universities. The findings reveal significant variability and deficiencies in several key areas: coverage of genetics and genomics, allocation of ECTS credits, teaching methodologies, lack of curriculum standardization, and the distribution of content between compulsory and elective subjects. These results have important implications for the preparation of nursing professionals in the era of personalized medicine.

### 4.1. Variability in the Inclusion of Genetics and Genomics

Of the 118 university centers analyzed, 10% (12 universities) does not include any reference to genetics and genomics in their teaching guides. This absence is particularly concerning given the growing relevance of these disciplines in diagnosis, treatment, and prevention of diseases, especially in the context of personalized medicine [16,20]. The lack of genetics and genomics education in a significant proportion of educational centers may impede the ability of future nurses to adequately perform functions such as genetic counseling and the clinical application of genomic knowledge.

Among the centers that do include genetic content, there is marked variability in the depth and focus of the topics taught. Approximately 43% of universities offering genetics limit themselves to providing basic training, addressing only the fundamentals of genetics. This introductory level may be insufficient for the complex challenges faced by nursing in managing patients with genetic pathologies, given the rapid advances in genomics in modern medicine [18,22,30]. In contrast, 57% of centers addresses more specialized topics, such as genetic counseling, pharmacogenetics, and other advanced clinical applications. However, these contents are often delivered theoretically and, in many cases, within elective courses, which may limit access to robust and consistent training in these areas for all students [7,31].

These findings align with studies conducted in other countries. In Latin America, research [32] has shown that the teaching of genetics and genomics in health professions often emphasizes theoretical content related to molecular and cellular bases, with a limited focus on clinical application. Similarly, in Asia, study [33] indicates that, while a significant proportion of molecular genetics content is included in nursing curricula, topics such as genetic counseling are less frequently covered. This suggests a global trend in which nursing education may not sufficiently address the practical and clinical aspects of genetics and genomics [14].

*4.2. Impact of Integrated vs. Stand-Alone Courses on Learning Outcomes*

An important consideration emerging from the findings is the approach to curriculum design—specifically, the integration of genetics and genomics content into broader modules versus offering dedicated, stand-alone courses. The observation that genetics and genomics topics are often integrated into broader subjects and do not receive dedicated focus suggests potential shortcomings in this teaching approach [30].

Educational literature indicates that, while integrated curricula can facilitate the contextual application of knowledge, they may not provide sufficient depth for complex subjects like genetics and genomics. Several studies have found that nursing students who participated in stand-alone genetics courses demonstrated higher competency levels, better knowledge retention, and increased confidence in applying genomic knowledge compared to those who received integrated instruction, suggesting that dedicated courses are more effective in developing the essential competencies required for genomic healthcare [13,34–37].

Our findings align with these studies, suggesting that the integration of genetics and genomics into broader modules may lead to superficial coverage, hindering the development of critical skills, such as genetic risk assessment and counseling [7,21,36,38,39]. This is further supported by the evidence from surveys and qualitative studies [4–8,20,37,40] reporting that nurses often feel unprepared to incorporate genomics into practice due to inadequate education, highlighting the need for comprehensive, stand-alone courses.

*4.3. Limited Allocation of ECTS Credits*

This study highlights the limited allocation of ECTS credits to genetics and genomics content. Nursing students generally complete between three and six ECTS in genetics and genomics, representing less than 2.5% of the total 240 ECTS credits required for the degree. This low academic load reflects a training gap in relation to the growing demand for genetic knowledge in the field of nursing [38]. With the advancement of personalized medicine, nurses are expected not only to understand basic genetics but also to apply this knowledge in daily practice, which requires more extensive and in-depth training [39].

The inadequate allocation of credits may result from crowded curricula and competing educational priorities. However, given the critical role genetics and genomics play in modern healthcare, it is imperative to re-evaluate and expand their inclusion in nursing programs [27–31].

Given the insufficient current allocation, it is pertinent to consider whether there should be a recommended number of ECTS credits dedicated to genetics and genomics in nursing curricula. International guidelines, such as those from the European Society of Human Genetics (ESHG) and the Global Genomics Nursing Alliance (G2NA), emphasize the necessity for comprehensive genetics/genomics education, but do not specify exact credit allocations [1,2,13,14].

Based on the complexity and importance of genetics and genomics in healthcare, we propose that nursing programs should allocate at least 10–12 ECTS credits (approximately 4–5% of the total degree credits) to these subjects. This increase would allow for a more thorough exploration of both theoretical concepts and practical applications, including genetic counseling, pharmacogenomics, and ethical considerations.

However, simply increasing the number of ECTS credits may not suffice to enhance educational quality. The effectiveness of genetics and genomics education also depends on:

- Teaching Methodologies: Active learning strategies, such as simulations, case studies, and laboratory work, are essential for developing practical competencies [40–42].
- Faculty Expertise: Educators must be adequately trained and confident in teaching genetics/genomics [15,20,42,43].
- Curriculum Integration: Genetics and genomics should be coherently integrated throughout the curriculum rather than being isolated in stand-alone courses [31–34].
- Practical Experience: Clinical placements that offer exposure to genetic counseling and genomic technologies can enhance learning outcomes [37,38,44].

Studies have shown that quality education results from a combination of sufficient instructional time and effective teaching practices [39,40,45]. Therefore, while increasing ECTS credits is a necessary step, it must be accompanied by curricular reforms and faculty development to truly improve genetics and genomics education in nursing.

### 4.4. Predominance of Theoretical Teaching Methodologies

The predominant educational methodology, centered on lectures and seminars, poses a challenge to the development of practical competencies essential for the clinical application of genetics and genomics. While this didactic approach is effective for transmitting theoretical knowledge, it does not sufficiently promote the development of skills, such as interpreting genetic tests, making clinical decisions based on genetic information, and providing genetic counseling—critical abilities in modern nursing practice [2,37,46].

Less than one-third of universities that teach advanced genetics content includes practical activities, such as clinical simulations, analysis of real cases, or work in specialized laboratories. The lack of practical training limits students' ability to apply what they have learned and may hinder their preparedness to integrate genetics and genomics into patient care effectively [12,13,30].

Research has shown that incorporating practical activities into genetics and genomics education significantly enhances students' ability to apply theoretical knowledge in clinical settings. Active learning strategies, such as simulations, case studies, and hands-on laboratory work, promote critical thinking and improve competence in genetic risk assessment and patient counseling [17,34,46].

Therefore, to enhance the preparedness of nursing graduates, it is crucial for nursing programs to incorporate practical components into genetics and genomics education. This could involve the use of simulation laboratories, collaborative projects with genetic counseling departments, or clinical placements focused on genomic healthcare [14,16,19].

### 4.5. Lack of Harmonization in Curricula

One of the most concerning findings is the lack of harmonization in nursing curricula across different universities. There is no national minimum standard that guarantees uniform competence in genetics and genomics for all nursing graduates in Spain. This disparity suggests that, depending on the university from which they graduate, nurses may have different levels of preparation to face genetic challenges in their professional practice [13,26,27].

Additionally, this variability extends within individual universities that have multiple campuses or affiliated centers offering nursing programs. Although some universities in Spain have curriculum committees or oversight structures aimed at ensuring consistency in content delivery across sites, our findings suggest that these mechanisms may not fully standardize the integration of genetics and genomics. This inconsistency highlights the need to strengthen institutional oversight and possibly implement national guidelines to ensure uniform content coverage, regardless of campus location [2,14,37].

The absence of a standardized curriculum poses a significant barrier to ensuring that all professionals have the capacity to offer high-quality healthcare based on genetic evidence. Establishing national competencies and integrating genetics and genomics consistently across nursing programs is essential to address this gap [20,30,31].

The successful standardization of other core subjects in the nursing curriculum demonstrates that harmonization is achievable. Applying similar strategies to genetics and genomics—such as developing national competencies, integrating standardized content, and enforcing accreditation requirements—could promote uniformity in education across institutions [13,36,38]. Moreover, emphasizing the assessment of genetic and genomic knowledge and skills is crucial. Educational literature underscores that assessment is a key driver of student learning. By implementing explicit assessment methods, such as practical evaluations, case studies, and competency-based assessments, educators can reinforce the

importance of genetics and genomics in nursing practice and encourage deeper student engagement [18,19,44].

### 4.6. Distribution of Content Between Compulsory and Elective Subjects

This study also reveals that the distribution of genetics and genomics content between compulsory and elective subjects may limit access to advanced genetics training. In many universities, basic genetic concepts are integrated into core subjects, such as Biology, Physiology, or Biochemistry, while more specialized content is offered in elective courses. This means that not all students may receive comprehensive training in genetics and genomics, as access depends on course availability, student choices, and university resources [12,14,42].

The fragmentation of advanced genetics teaching across various subjects and its presentation as a marginal complement rather than an integrated area within nursing may prevent students from perceiving its relevance. Integrating genetics and genomics more cohesively into the core curriculum could enhance the preparedness of nursing graduates [17,30,38].

Improving the integration of genetics and genomics into the curriculum has been highlighted as a key strategy to prepare future nursing professionals. This requires shifting from fragmented teaching approaches to a unified framework that promotes equal access and emphasizes the clinical importance of genetics and genomics in nursing practice [22,31,37].

### 4.7. Global Efforts to Enhance Genetics and Genomics Education

It is noteworthy that, since the 1990s, efforts have been made to better prepare nurses in genetics and genomics by influential organizations, such as the American Association of Colleges of Nursing (AACN), the American Nurses Association, the American Academy of Nursing, and the International Society of Nurses in Genetics. Each has published position statements and competencies to guide the integration of genetics and genomics into nursing education [30,34,37,47].

Similar initiatives are needed in Spain to develop national competencies and guidelines that can harmonize nursing education in genetics and genomics. By adopting international best practices and tailoring them to the Spanish context, nursing programs can ensure that graduates are equipped to meet the demands of modern healthcare [2,13,14,22,46].

### 4.8. Implications for Nursing Practice and Patient Care

The partial and unequal integration of genetics and genomics in nursing programs has significant implications for patient care. Nurses are often the first point of contact in healthcare settings and play a crucial role in patient education, risk assessment, and coordination of care [9,13,20]. Without adequate training, nurses may be ill-prepared to address the complexities of genetic and genomic information, potentially impacting patient outcomes [31,44].

In the context of precision medicine, which tailors medical treatments based on patients' genetic, environmental, and lifestyle factors, the need for nurses to have a solid foundation in genetics and genomics becomes even more critical. Genomic education is essential for nurses to accurately interpret genetic test results, identify potential risks, and personalize care plans. As seen in regions such as Indonesia or Serbia, integrating genomic nursing into educational curricula equips professionals to actively participate in interdisciplinary teams that provide care rooted in patient-specific genetic information, optimizing clinical decision making and improving health outcomes [23,39,42].

The integration of genetics and genomics into clinical nursing practice represents a pivotal advancement toward personalized healthcare, enabling tailored treatments and preventive strategies based on an individual's genetic profile. For example, in oncology, genetic profiling allows for the selection of targeted therapies, improving treatment efficacy and reducing side effects [16,20]. Similarly, in cardiology, genomics informs risk

assessments and guides interventions for hereditary heart conditions, facilitating proactive management and prevention [36,47].

Nurses play a crucial role by educating patients about genetic risks, interpreting genetic test results, and coordinating care plans that incorporate genetic information. This personalized approach not only enhances the quality of care, but also empowers patients to make informed health decisions [10,31]. However, in Spain, the lack of a formal regulatory framework for genetic counseling limits nurses' involvement, confining their role to foundational education and emotional support without specialized focus [5,14]. Addressing this gap through standardized education and clear role definitions is essential for maximizing the potential of nursing contributions to genetic and genomic healthcare [8,46].

Enhancing genetics and genomics education can empower nurses to contribute effectively to interdisciplinary teams, engage in genetic counseling, and participate in the implementation of personalized medicine strategies [13,19,37,38]. In Spain, this educational improvement will not only allow nurses to identify genetic risks more effectively, but also enable them to play a pivotal role in designing prevention strategies and personalized treatment plans, thereby reinforcing the future of healthcare delivery.

*4.9. Recommendations for Curriculum Improvement*

Given the findings of this study, several recommendations emerge:

1.  Development of National Competency Framework: Establishing standardized competencies for genetics and genomics can ensure consistent education across nursing programs [2,13,14,38,44]. Drawing on models from other countries, Spain can develop guidelines tailored to its educational context
    Proposed Mechanism:

    *   Collaboration Between Government and Educational Institutions: The Ministry of Education and the Ministry of Health can jointly establish a task force comprising representatives from universities, nursing faculties, professional nursing organizations, and genetics/genomics experts. This task force would be responsible for developing national competencies and standardized curricula for genetics and genomics education in nursing.
    *   Engagement of Professional Nursing Organizations: Organizations such as the General Council of Nursing in Spain can advocate for the development of these competencies and facilitate communication between educators and policymakers.

2.  Integration of Genetics and Genomics Throughout the Curriculum: Embedding these topics into compulsory courses and across various subjects can reinforce their relevance to nursing practice [17,19–21,34,46,48].
    Proposed Mechanism:

    *   Curriculum Development Workshops: Organize national workshops involving curriculum developers, faculty, and experts to design integrated curriculum models.
    *   Pilot Programs: Implement pilot integrated curricula in select nursing schools to evaluate effectiveness before wider adoption.

3.  Increase in ECTS Credits Allocation: Allocating more credits to genetics and genomics will provide students with sufficient time to develop essential competencies [6,22,26,27,36].
    Proposed Mechanism:

    *   Policy Advocacy and Legislation: Stakeholders can lobby for regulatory changes that mandate a minimum number of ECTS credits for genetics and genomics in nursing programs.
    *   Collaboration with Accreditation Bodies: Work with accreditation agencies to include genetics and genomics credit requirements in accreditation standards.

4. Enhancement of Teaching Methodologies: Incorporating active learning strategies, such as simulations, case studies, and laboratory work, can improve engagement and practical skill development [20,39,42,44,46]. Specific methods include clinical simulations for practicing genetic counseling, laboratory workshops on DNA testing, and interdisciplinary case studies to foster collaborative learning with other healthcare disciplines. Virtual tools and self-paced e-learning modules should also be employed to complement traditional teaching methods.
   Proposed Mechanism:

   - Faculty Development Programs: Universities can partner with genetic research centers and international organizations to provide training and resources for faculty on active learning methodologies.
   - Funding Opportunities: Seek grants from government agencies or private foundations to support the development of simulation labs and acquisition of teaching resources.

5. Faculty Development: Investing in training for educators can improve the quality of genetics and genomics instruction [3,13,30,31,37,45,47,48].
   Proposed Mechanism:

   - Professional Development Initiatives: Establish programs to enhance faculty expertise in genetics and genomics, including workshops, continuing education courses, and collaborations with genetic specialists.
   - International Collaboration: Engage with global initiatives and learn from countries that have successfully integrated genetics and genomics into nursing education to adopt best practices.

Implementing these recommendations requires a coordinated effort among key stakeholders:

- Government Agencies: The Ministry of Education and Ministry of Health play pivotal roles in policy development and regulatory changes.
- Professional Nursing Organizations: Can advocate for curriculum changes, facilitate communication among stakeholders, and support policy initiatives.
- Educational Institutions: Universities and nursing schools can participate in pilot programs, faculty development, and curriculum integration efforts.
- Faculty and Educators: Involvement in curriculum design, adopting new teaching methodologies, and participating in professional development.

While these recommendations are tailored to the Spanish context, many countries face similar challenges in integrating genetics and genomics into nursing education. The proposed mechanisms can serve as a framework for other nations. By fostering international collaboration and sharing best practices, the global nursing community can work toward enhancing genetics and genomics competencies worldwide.

### 4.10. Study Limitations

One potential limitation of this study is related to the timeliness of the information collected. Teaching guides are dynamic documents that may undergo revisions and updates, particularly in response to curriculum changes or institutional policies. Although the study utilized the most up-to-date guides available during the review period, it is possible that some content was modified after data collection, potentially impacting the accuracy of the findings in reflecting the most current genetics and genomics curriculum across institutions.

Another limitation pertains to the variability in course nomenclature across institutions, which posed challenges during the data standardization process. Different universities employed diverse terminologies and descriptions for courses covering similar topics, necessitating meticulous classification and categorization of content. This process required rigorous scrutiny to achieve consistency and accuracy in data categorization, enabling valid comparisons across institutions. Despite these efforts to standardize the information, the

inherent variability in course nomenclature may still introduce some limitations in the direct comparability of course content across different programs.

Together, these factors underscore the need for ongoing studies that monitor curriculum changes over time and consider standardized nomenclature frameworks to improve the consistency of course descriptions. Future research should also focus on establishing national guidelines for genetics and genomics terminology within nursing education to enhance cross-institutional comparability and alignment with evolving healthcare demands.

## 5. Conclusions

With approximately 5.9 nurses per 1000 inhabitants—below the European average of 8.8—Spain faces a nursing shortage that may impact the integration of genetics education into nursing curricula [49]. Despite thousands of nursing students graduating annually, the depth and extent of genetics and genomics education in their programs remain unclear. This limited visibility into genetics education may restrict the nursing profession's capacity to expand its role in this field, ultimately affecting nurses' preparedness to meet the genetic and genomic needs of modern healthcare.

The findings of this study highlight the imperative need for structural improvements in genetics and genomics training within Spanish nursing programs. The partial and unequal integration of these critical areas, characterized by limited thematic coverage, an inadequate allocation of ECTS credits, and a lack of practical focus in teaching methodologies, undermines the preparedness of nursing graduates to apply genetic advances in healthcare. Establishing a national minimum standard for genetics and genomics education is essential to ensure that all future nurses are equipped to address the clinical challenges posed by genetic pathologies, thereby contributing to enhanced healthcare outcomes in the 21st century.

Moreover, this study underscores the importance of introducing a dedicated subject or module on genetics and genomics within nursing curricula. Without a focused approach, there is a significant risk that these essential topics will remain peripheral to other health and nursing subjects, limiting their impact on student learning and professional readiness. A stand-alone course would provide the necessary depth and consistency, ensuring future nurses develop a solid foundation in genetic and genomic competencies. This dedicated training is crucial for enabling nurses to effectively integrate genetic information into patient care, ultimately enhancing personalized healthcare outcomes.

By reviewing and updating nursing curricula, increasing the allocation of ECTS credits to genetics and genomics, enhancing teaching methodologies, and fostering faculty development, nursing education in Spain can be aligned with the evolving demands of healthcare. This alignment will not only improve the quality of nursing education, but also enhance patient care, as nurses play a vital role in integrating genomics into healthcare delivery. Implementing standardized and comprehensive genetics and genomics education is essential for meeting the challenges of modern healthcare and ensuring high-quality, personalized patient care in the genomic era.

**Author Contributions:** Conceptualization, E.R.-B., C.G.-S. and D.G.-G.; methodology, E.R.-B., C.G.-S., D.G.-G. and K.A.-M.; software, E.R.-B., C.G.-S. and D.G.-G.; validation, E.R.-B., C.G.-S. and D.G.-G.; formal analysis, E.R.-B., C.G.-S., D.G.-G. and K.A.-M.; investigation, E.R.-B., C.G.-S., D.G.-G. and K.A.-M.; resources, E.R.-B., C.G.-S. and D.G.-G.; data curation, E.R.-B., C.G.-S. and D.G.-G.; writing—original draft preparation, E.R.-B., C.G.-S., D.G.-G. and K.A.-M.; writing—review and editing, E.R.-B., C.G.-S. and D.G.-G.; visualization, E.R.-B., C.G.-S. and D.G.-G.; supervision, E.R.-B., C.G.-S. and D.G.-G. All authors have read and agreed to the published version of the manuscript.

**Funding:** This research received no external funding.

**Institutional Review Board Statement:** Not applicable.

**Informed Consent Statement:** Not applicable.

**Data Availability Statement:** No new data were generated or analyzed in this study.

**Public Involvement Statement:** There was no public involvement in any aspect of this research.

**Guidelines and Standards Statement:** This manuscript was drafted following the PRISMA guidelines for systematic reviews and meta-analyses.

**Use of Artificial Intelligence:** AI or AI-assisted tools were not used for drafting any aspect of this manuscript.

**Conflicts of Interest:** The authors declare no conflicts of interest.

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
