# Peer review of "Analysis of Nursing Education Curricula in Spain: Integration of Genetic and Genomic Concepts"

_nursrep, doi:10.3390/nursrep14040270_

Round 1

Reviewer 1 Report

Comments and Suggestions for Authors

The article by Ramirez-Baraldes et al, is well addresed and written and its mainly focused in to assess the current state of genetic and genomic training in nursing degree programs across Spanish universities however  I have some comments:

First of all,  it said in the abstract “The integration of genetics and genomics into nursing practice is essential for addressing genetic pathologies and providing personalized patient care”. After Reading the paper it is not clear for me the comment aboved. I wonder the autor must explore more this aspect. The do not discuss especific aspects of the Genetics in Spain. The current situation of the geneticist in Spain is not formaly regulated, so it is not surprised the data regarding training of nurses in genetic and genomics in Spain. We have to ask…. If the situation is similar in some other countries maybe is because it is not the main ítem to be trained in nursing. That aspect has to be discussed more profusely. I can not be compare with USA an the training programs in this country.

Sentences such as: The integration of genetics and genomics into clinical nursing practice represents a pivotal advancement toward personalized healthcare, enabling tailored treatments and preventive strategies based on an individual's genetic profile. Has to be deeply discussed.

Regarding the results they are interesting but I will expect that because genetic and genomic training are not in the main competencies of Nurses. I will agreed with the authors that better trainies or eduction in genetic for nurses are great but other genetic and genomic profiles in Spain are prepared are trained for that, such as genetic counsellors and clinical geneticists that aspect must be discussed.

Other aspects:

-Introduction is too large.

-Part of the discussion repeat data results it must be discussed.

- I really recommend the aspect of the recommendations in the discussion and limitations of the work-

- A table  with all result generated may help to the reader.

Author Response

Dear Reviewer,

We sincerely appreciate your constructive and detailed comments on our manuscript titled "Analysis of Nursing Education Curricula in Spain: Integration of Genetic and Genomic Concepts." Your observations have provided us with valuable insights to enhance the quality and clarity of our work.

Below, we address each of your comments and describe the modifications we have made or will make in the manuscript.

The article by Ramirez-Baraldes et al, is well addresed and written and its mainly focused in to assess the current state of genetic and genomic training in nursing degree programs across Spanish universities however  I have some comments:

First of all,  it said in the abstract “The integration of genetics and genomics into nursing practice is essential for addressing genetic pathologies and providing personalized patient care”. After Reading the paper it is not clear for me the comment aboved. I wonder the autor must explore more this aspect. The do not discuss especific aspects of the Genetics in Spain. The current situation of the geneticist in Spain is not formaly regulated, so it is not surprised the data regarding training of nurses in genetic and genomics in Spain. We have to ask…. If the situation is similar in some other countries maybe is because it is not the main ítem to be trained in nursing. That aspect has to be discussed more profusely. I can not be compare with USA an the training programs in this country.

Thank you for your thoughtful and constructive feedback on our manuscript.

  1. Clarification of Abstract Statement: We have expanded the manuscript to provide a clearer explanation of how integrating genetics and genomics into nursing practice is essential for addressing genetic pathologies and facilitating personalized patient care in Spain.
  2. Specific Aspects of Genetics in Spain: We have included a detailed discussion on the current regulatory framework for genetic counseling in Spain, highlighting the absence of formal regulation and its implications for nursing education in genetics and genomics.
  3. International Context: We have elaborated on why genetics may not be a primary focus in nursing training in Spain and similar countries, without directly comparing it to the USA. This section explores broader international trends and the factors influencing the prioritization of genetics in nursing curricula.

These revisions enhance the clarity and depth of our manuscript, addressing the areas you identified. We believe these changes provide a more comprehensive understanding of the integration challenges and underscore the importance of standardized genetics and genomics education in nursing.

Sentences such as: The integration of genetics and genomics into clinical nursing practice represents a pivotal advancement toward personalized healthcare, enabling tailored treatments and preventive strategies based on an individual's genetic profile. Has to be deeply discussed.

Thank you for your valuable feedback on our manuscript. We appreciate your suggestion to deepen the discussion on the integration of genetics and genomics into clinical nursing practice, particularly within the Spanish context.

In response, we have enhanced the introduction to more clearly articulate the importance of integrating genetics and genomics for personalized healthcare in Spain. Additionally, we have expanded Section 4.8, "Implications for Nursing Practice and Patient Care," by adding a dedicated subsection that thoroughly explores the significance, challenges, and implications of incorporating genetics and genomics into nursing practice in Spain.

These revisions provide a more comprehensive analysis of the role of genetics and genomics in nursing, effectively addressing the concerns you raised.

Regarding the results they are interesting but I will expect that because genetic and genomic training are not in the main competencies of Nurses. I will agreed with the authors that better trainies or eduction in genetic for nurses are great but other genetic and genomic profiles in Spain are prepared are trained for that, such as genetic counsellors and clinical geneticists that aspect must be discussed.

We acknowledge and appreciate your observation that genetic and genomic training has not traditionally been a primary competency within nursing education. To address your suggestion, we have expanded the Discussion section to include a detailed examination of the roles of genetic counsellors and clinical geneticists in Spain and their interplay with nursing competencies. The following additions have been made:

  1. Roles and Responsibilities:
    • Genetic Counsellors and Clinical Geneticists: We have included a comprehensive description of the functions of genetic counsellors and clinical geneticists within the Spanish healthcare system. These professionals are primarily responsible for conducting genetic assessments, providing specialized counseling, and interpreting complex genetic information.
    • Nurses: The expanded discussion highlights the complementary role of nurses, who provide foundational genetic education, initial risk assessments, and emotional support to patients and families. While nurses may not possess the same depth of specialized training as genetic counsellors, their involvement is crucial in facilitating patient care and bridging gaps in genetic services.
  2. Interprofessional Collaboration:
    • We have emphasized the importance of collaboration between nurses and specialized genetic professionals. This collaboration ensures comprehensive patient care, where nurses prepare patients for genetic testing, assist in sample collection, and facilitate follow-up care based on genetic findings.
    • The manuscript now discusses how effective communication and coordination between these roles enhance the integration of genetic information into patient care plans, thereby improving personalized medicine outcomes.
  3. Importance of Nursing Education in Genetics and Genomics:
    • The discussion now underscores the necessity for nurses to possess adequate genetic and genomic knowledge to support and complement the work of genetic counsellors and clinical geneticists. This is particularly important in settings where access to specialized genetic professionals may be limited.
    • We argue that enhanced training in genetics and genomics empowers nurses to advocate for patients, educate them about genetic risks, and assist in navigating genetic information, thereby playing a pivotal role in patient-centered care.
  4. Regulatory and Educational Frameworks:
    • We have included an analysis of the current regulatory framework in Spain, highlighting the need for formal recognition and support of genetic roles within nursing. This includes recommendations for establishing clear guidelines and competencies to facilitate better integration and collaboration between nurses and genetic professionals.
    • Additionally, the manuscript now discusses educational reforms necessary to incorporate genetics and genomics into nursing curricula, ensuring that nurses are adequately prepared to engage with genetic information in their practice.

These additions provide a more comprehensive view of the genetic and genomic education landscape for nurses in Spain, addressing the interaction and complementarity between nursing and specialized genetic professions. By elucidating these relationships, the manuscript now offers a holistic understanding of how nursing education can evolve to meet the growing demands of genetic and genomic healthcare.

Other aspects:

-Introduction is too large.

We acknowledge your comment regarding the length of the Introduction section. To provide a comprehensive understanding of the Spanish context and the specific role of nurses in genetic and genomic healthcare, we have streamlined the Introduction while ensuring that essential background information is retained. This adjustment maintains the necessary depth to justify the study's focus and relevance without compromising conciseness.

-Part of the discussion repeat data results it must be discussed.

Thank you for your insightful comment. We have revised the discussion section to minimize the repetition of specific results data. Changes have been made throughout various parts of the text to focus more on the interpretation of findings and their implications, rather than reiterating specific figures or details already presented in the results section. The discussion now emphasizes analysis and practical recommendations for improving genetics and genomics content within nursing education in Spain.

- I really recommend the aspect of the recommendations in the discussion and limitations of the work-

Thank you for your thoughtful recommendation to expand on specific recommendations within the discussion and limitations sections. We appreciate your feedback and are pleased to confirm that these aspects are comprehensively addressed in the manuscript. The discussion section includes detailed recommendations, emphasizing curriculum reforms, the establishment of a national competency framework, increased ECTS credit allocation, active teaching methodologies, and faculty development. Additionally, the limitations section provides a reflection on potential constraints of the study and suggests avenues for future research that assess the implementation and outcomes of these recommendations. We believe these components collectively address the points you raised, ensuring a rigorous and practical approach to our findings.

- A table with all result generated may help to the reader.

Thank you for the excellent suggestion. We agree that a table compiling all results would be very helpful for the reader. However, due to the large number of teaching plans analyzed, such a table would be extensive, spanning multiple pages. Given the space limitations for this publication, including such a detailed table is unfortunately not feasible. We have, however, aimed to summarize the key findings in the text to convey the most important insights effectively.

Thank you once again for your constructive feedback and for contributing to the improvement of our manuscript.

Sincerely,

Reviewer 2 Report

Comments and Suggestions for Authors

The paper is very nicely formulated and the results are systematically presented.

Is there any data on the number of students per faculty and what percentage of them do not receive any training (genetics and genomics are not on the curriculum)?

The method of conducting the study (via PRIZMA guides) is mentioned in the methodology.

Is there perhaps a specific subject with this name or only with this content in any curriculum?

Limitation: is there any data on how many courses (faculties) were not available for analysis.

Briefly suggest the content, topics and learning methods that would be appropriate for the curriculum in this subject.

Conclusion: clearly emphasize the need for a dedicated genetics and genomics subject as information will not be taught if it is peripheral to health and nursing topics.

Author Response

Dear Reviewer,

We sincerely appreciate your constructive and detailed comments on our manuscript titled "Analysis of Nursing Education Curricula in Spain: Integration of Genetic and Genomic Concepts." Your observations have provided us with valuable insights to enhance the quality and clarity of our work.

Below, we address each of your comments and describe the modifications we have made or will make in the manuscript.

The paper is very nicely formulated and the results are systematically presented.

Is there any data on the number of students per faculty and what percentage of them do not receive any training (genetics and genomics are not on the curriculum)?

Thank you for your insightful feedback and positive remarks on our manuscript. We appreciate your suggestion to include data on the number of nursing students impacted by the absence of genetics and genomics in curricula. In response, we have added information in the introduction on the approximate number of nursing graduates in Spain to provide context on the scale of professionals potentially affected by curricular gaps.

Regarding the percentage of students who may not receive genetics and genomics training, we addressed this indirectly by analyzing the extent to which these topics are covered in compulsory versus elective courses. This approach reflects our focus on curricular content structure, indicating the level of mandatory versus optional exposure to these critical areas. Thank you for prompting us to strengthen these aspects of the study.

The method of conducting the study (via PRIZMA guides) is mentioned in the methodology.

Thank you for your observation. To clarify, while this study focused on reviewing teaching guides rather than research articles, as is common in PRISMA-guided reviews, we chose to follow the PRISMA framework as an overarching structure. This approach was adopted to ensure scientific rigor and transparency in our methodology, aligning with best practices for systematic reviews. By adhering to PRISMA’s structured approach, our goal was to maintain the highest level of academic and scientific rigor in evaluating nursing curricula across Spanish universities. We trust that this clarification underscores our commitment to methodological.

Is there perhaps a specific subject with this name or only with this content in any curriculum?

Thank you for your valuable question and insightful comments. In response, we have included specific information on the presence of genetics and genomics courses within Spanish nursing curricula. While dedicated courses exclusively focused on "Genetics and Genomics" are rare, some universities offer individual courses or sections within broader subjects that cover these topics. Typically, genetics and genomics content is integrated into courses such as general biology, molecular biology, or specialized electives. This approach generally provides a foundational overview, although it may not always allow for an in-depth exploration of advanced topics such as genetic counseling or pharmacogenomics.

To better illustrate this variation, we have added examples of course titles and content focus in section 3.6, detailing courses from basic to more specialized levels. This addition provides a clearer and more comprehensive view of how these topics are integrated across different nursing programs. We appreciate your feedback, which has helped enhance the academic rigor and clarity of our study.

Limitation: is there any data on how many courses (faculties) were not available for analysis.

Thank you for your insightful question. We have addressed this point in the manuscript, noting that all publicly accessible teaching guides from the 118 university centers were included in our review. Of these, we identified 12 centers without any genetics or genomics content, which were subsequently excluded from the main analysis. Minor accessibility issues encountered with a few specific guides were resolved to ensure comprehensive data inclusion.

We appreciate your input, which has contributed to clarifying these aspects within the limitations section of the manuscript.

Briefly suggest the content, topics and learning methods that would be appropriate for the curriculum in this subject.

Thank you for your insightful suggestion. In response to your comment, we have ensured that the manuscript now explicitly addresses appropriate learning methods for genetics and genomics education within nursing curricula. Specifically, we highlight the importance of incorporating active learning strategies, such as clinical simulations for practicing genetic counseling, laboratory workshops focused on DNA testing, and interdisciplinary case studies to promote collaborative learning with other healthcare disciplines. Additionally, we emphasize the use of virtual tools and self-paced e-learning modules to complement traditional teaching methods and enhance accessibility and engagement. These revisions aim to provide a comprehensive and practical approach to improving nursing education in this critical area.

Conclusion: clearly emphasize the need for a dedicated genetics and genomics subject as information will not be taught if it is peripheral to health and nursing topics.

Thank you for your valuable feedback regarding the emphasis on a dedicated genetics and genomics subject within nursing curricula. In response to your suggestion, we have revised the conclusions to highlight the critical need for a standalone module in genetics and genomics. This modification underscores the importance of integrating these essential topics more thoroughly into nursing education, ensuring that they are not treated as peripheral subjects. By doing so, we aim to enhance the professional preparedness of nursing graduates to address the genetic and genomic challenges in modern healthcare effectively.

Thank you again for your insightful comments, which have significantly enhanced the quality of our paper.

Sincerely,

Reviewer 3 Report

Comments and Suggestions for Authors

This paper, based on a review of Spanish curricula documentation (2023-2024), describes the inclusion of genetics and genomics in pre-qualification nursing courses.

Given the growing number of genomic healthcare applications and the recognition internationally, of the role that nurses will play in delivering genomic informed care, it is important to understand how the health workforce (including nurses) are prepared in relation to this topic. This is a useful paper that provides country-specific insights which has relevance to nursing education and genomics education in other countries.

As a first step, the authors need to review all references (both in-text and reference list) as a number of errors have been identified. At this time, this reviewer is unable to sufficiently review the paper to confirm whether the references used are appropriate.

The following are examples of the errors identified (it is not an exhaustive list). Errors include:

·       incorrect numbering - where numbers in the text do not relate to the reference in the bibliography eg ref 7 (line 65) and ref 8 (line 67) are 10 and 11 in the reference list

·       papers do not appear to exist eg ref 7: Anderson, P. L., & Harris, K. (2019). Personalized medicine and nursing education: Preparing the workforce for genomic advancements. Journal of Nursing Education, 58(3), 135-142. https://doi.org/10.3928/01484834-20190220-04  I have searched separately for the title, by the two authors and using the doi (which is ‘not found’)

·       hyperlinks in the bibliography are incorrect and takes a reviewer to a different reference eg reference 33 goes to the correct journal (Nursing Management) but it is to the paper Reflective practices: meaningful recognition for healthy work environments

Similarly reference 34:  https://doi.org/10.1111/j.1547-5069.2011.01385.x  goes to a JNS paper but it is Accuracy and Efficiency of Novice Nurse Practitioners Using Personal Digital Assistants

·       at least one paper (#34 in the reference list) has not been written by one of the cited authors and this reviewer is unable to find a paper with that title.

Key points that should be addressed to strengthen the manuscript:

Introduction:

Inclusion of information that will provide context for this specific study would be welcomed by the international readership who may know little about Spain. This could include:

·       a brief description of genetic/genomic services currently available in Spain and whether this is usually provided in specialist services (eg by doctors specialising in genetics/genomics) or whether testing is offered in routine care settings.

·       existence or absence of genetic counselling as a recognised profession and whether the presence/absence of this role has any impact on what nurses could/should do

·       current scope of Spanish nursing practice especially in relation to genetics/genomics [paragraph beginning line 53 refers to nurses in oncology and cardiology, but given the issues with references (detailed below) it is unclear whether this refers to Spain or elsewhere]

·       number of nurses in Spain; are their enough; approx. how many are being trained across all of the universities each year. [This may provide context to the impact of 10% of sites not teaching genetics/genomics]

·       what are ‘the evolving demands of healthcare’ in Spain, particularly around genomics and whether there are local and/or national strategies in genomics driving change.

Genetic counselling – introduced on page 2 and referred to in a number of places throughout the manuscript. As an activity provided by nurses, the genetic counselling that the authors refer to should be briefly described and discussed in terms of how this may be similar, different to, or overlapping with the genetic counselling provided by individuals formally recognised as genetic counsellors due to completion of extensive training and registration processes.

Line 64. A rephrasing of this sentence is needed as G2NA has not developed ‘competencies’ in integration and leadership

Line 96 more information is needed on the ‘expert nurse in genetics and genomics’ and how establishing this role will help address education of nursing graduates (as described in the sentence before).

Materials and Methods

If the predefined evaluation matrix (line 164) is different to the data extraction information (line 168-175) please provide details of what this matrix contains. If they are the same thing, please be consistent in your terminology.

Please provide details of the team’s interpretation of ‘basic vs advanced’ content (line 184).

In 3.1 examples of content (‘specialized areas that are crucial for clinical practice’ line 212) are provided. Did the authors start the study with curricula content that they expected to find (eg based on the nursing/genomics literature or established nursing competencies in genomics eg Eurogentest)? [If they did, this detail should be included in the Methods section. Absence of expected topics in curriculum document(s) should be included in the findings]

Results

If 118 centres align to 65 universities, how are the 12 that do not teach genetics/genomics distributed? Is it that a small number of universities don’t teach it across any of their affiliates and campuses, or are there up to 12 universities where one site does not include it? Line 273 suggests that it is the latter. Do universities have mechanisms to identify and deal with inconsistencies in teaching the same course across multiple sites?

Line 404 indicates that some institutions were excluded from the analysis. This detail and number of institutions needs to be included in the results section.

The inclusion of anonymised examples of how universities describe the content that will be covered in courses would be helpful to illustrate difference between universities.

Discussion

Much of the current discussion is a repetition of the findings. A more rigorous discussion of the results is needed and could draw on wider educational literature. For example:

-          line 215-216 ‘although these are usually integrated into broader modules and often do not receive dedicated focus.’ This suggests that integration may be a less effective teaching approach for genetics/genomics. What is the evidence around stand-alone vs integrated genetics/genomics content on student knowledge (or other outcomes?)

-          Line 221 calculation of genetics/genomics ECTS and the statement that currently the figure is insufficient, provides an opportunity to discuss whether there should be a recommended number of ECTS and what this number should be. Will quantity (of teaching) be a benchmark for quality?

-          Line 231-233 “without practical activities that allow students to develop applied competencies. This approach negatively impacts the development of practical skills that are essential for applying this knowledge in direct patient care.” Without evidence that skill development or application of knowledge is impacted at these Spanish universities, this sentence in the findings section should be written as an assumption rather than a statement of fact. The authors have an opportunity in the discussion to reference other approaches to teaching genetics/genomics that have been shown to be effective.

-          Section 3.4 standardisation. Are there other content areas of the Spanish nursing curricula where there is standardisation and if there are, how has this been achieved?

The reviewer is assuming that at least some of the curriculum documents would contain information on assessment. It would also be helpful to know if any made explicit reference to assessment of genetic/genomic knowledge and/or skills as assessment is often discussed as a driver for learning.

The authors make a number of recommendations (subsection 4.8) and call for the establishment of national minimum standards for genetics and genomics education (Conclusions line 431). The authors should propose a mechanism(s) as to how these recommendations might be achieved and outline, for example, the individuals, organisations and/or professional bodies that need to be involved to drive strategy and change policy. Given that many other countries are likely to be in a similar situation, readers may be interested to understand how these challenges could be tackled.

line 386 If more than one competency framework is required for Spanish nursing education, please provide more information. If this is a typographical error the ‘s’ should be removed from Frameworks

Author Response

Dear Reviewer,

We sincerely appreciate your constructive and detailed comments on our manuscript titled "Analysis of Nursing Education Curricula in Spain: Integration of Genetic and Genomic Concepts." Your observations have provided us with valuable insights to enhance the quality and clarity of our work.

Below, we address each of your comments and describe the modifications we have made or will make in the manuscript.

As a first step, the authors need to review all references (both in-text and reference list) as a number of errors have been identified. At this time, this reviewer is unable to sufficiently review the paper to confirm whether the references used are appropriate.

The following are examples of the errors identified (it is not an exhaustive list). Errors include:

  • incorrect numbering - where numbers in the text do not relate to the reference in the bibliography eg ref 7 (line 65) and ref 8 (line 67) are 10 and 11 in the reference list
  • papers do not appear to exist eg ref 7: Anderson, P. L., & Harris, K. (2019). Personalized medicine and nursing education: Preparing the workforce for genomic advancements. Journal of Nursing Education, 58(3), 135-142. https://doi.org/10.3928/01484834-20190220-04  I have searched separately for the title, by the two authors and using the doi (which is ‘not found’)
  • hyperlinks in the bibliography are incorrect and takes a reviewer to a different reference eg reference 33 goes to the correct journal (Nursing Management) but it is to the paper Reflective practices: meaningful recognition for healthy work environments

Similarly reference 34:  https://doi.org/10.1111/j.1547-5069.2011.01385.x  goes to a JNS paper but it is Accuracy and Efficiency of Novice Nurse Practitioners Using Personal Digital Assistants

  • at least one paper (#34 in the reference list) has not been written by one of the cited authors and this reviewer is unable to find a paper with that title.

Thank you for bringing these critical errors in our references to our attention. We have conducted a thorough review of all in-text citations and the reference list to address the issues you identified.

  • Numbering and Correspondence: We have adjusted the numbering of the references within the text to ensure they accurately correspond to the entries in the reference list. This should resolve any inconsistencies between in-text citations and the bibliography.
  • Verification of References: Each reference has been meticulously verified to confirm its existence and accuracy. In cases where errors were found in the citation details, such as incorrect author names, titles, or DOIs, we have corrected them and provided the correct information.
  • Correction of Hyperlinks: We have corrected the hyperlinks in the reference list to ensure they direct to the appropriate articles. All URLs and DOIs have been tested and are now fully functional.
  • Authorship and Content: We have reviewed all references to confirm that the authorship and content match the citations in the text and that they are relevant to our study.

We apologize for any inconvenience these errors may have caused and appreciate your diligence in highlighting them. We are confident that the revised manuscript now reflects accurate and appropriate references, enhancing the integrity and clarity of our work.

Key points that should be addressed to strengthen the manuscript:

Introduction:

Inclusion of information that will provide context for this specific study would be welcomed by the international readership who may know little about Spain. This could include:

  • a brief description of genetic/genomic services currently available in Spain and whether this is usually provided in specialist services (eg by doctors specialising in genetics/genomics) or whether testing is offered in routine care settings.
  • existence or absence of genetic counselling as a recognised profession and whether the presence/absence of this role has any impact on what nurses could/should do
  • current scope of Spanish nursing practice especially in relation to genetics/genomics [paragraph beginning line 53 refers to nurses in oncology and cardiology, but given the issues with references (detailed below) it is unclear whether this refers to Spain or elsewhere]
  • number of nurses in Spain; are their enough; approx. how many are being trained across all of the universities each year. [This may provide context to the impact of 10% of sites not teaching genetics/genomics]
  • what are ‘the evolving demands of healthcare’ in Spain, particularly around genomics and whether there are local and/or national strategies in genomics driving change.

 Thank you for your valuable feedback on the need to provide more context around genetic and genomic services in Spain for international readers unfamiliar with the region. We have carefully reviewed the introduction and incorporated detailed information to address each of the points raised:

  • Description of Genetic/Genomic Services: We have included a brief overview of the genetic and genomic services currently available in Spain, noting that these services are primarily provided in specialized centers and tertiary hospitals. This information highlights that genetic counseling and testing services are less common in routine primary care settings, which limits access for patients outside specialized care.
  • Existence of Genetic Counseling as a Profession: We have clarified that genetic counseling is not formally recognized as a separate profession in Spain. As a result, genetic counseling responsibilities typically fall to physicians with specialized genetics training, rather than dedicated genetic counselors. This absence of a defined genetic counseling role impacts the potential for nurses to engage in this work and underscores the need for further regulatory support to expand the scope of nursing practice in this field.
  • Scope of Nursing Practice in Relation to Genetics/Genomics: We have added information on the current scope of nursing practice in genetics and genomics in Spain, including a discussion of specialized roles in oncology and cardiology. This section explains that although some nurses in these fields may integrate genetic counseling into their practice, this level of involvement is limited and less common compared to other countries where nursing roles in genetics are more established.
  • Number of Nurses and Training Programs in Spain: To provide context on the capacity for integrating genetics education, we have included statistics indicating that Spain has approximately 5.9 nurses per 1,000 inhabitants, below the European average. Additionally, we note that while thousands of nursing students graduate each year, only a small percentage of programs currently offer genetics and genomics education, highlighting a significant gap in foundational training.
  • Evolving Demands in Spanish Healthcare: We have outlined the evolving demands in Spanish healthcare, particularly in the context of genomics, and referenced national strategies aimed at enhancing the integration of genomics into clinical care. These insights emphasize the growing importance of genomics in Spanish healthcare and the need for nursing curricula to adapt accordingly.

We appreciate your guidance in enhancing the introduction to provide a comprehensive background on the specific context of genetic and genomic services in Spain. We believe these additions make the manuscript more informative and accessible to an international audience.

Genetic counselling – introduced on page 2 and referred to in a number of places throughout the manuscript. As an activity provided by nurses, the genetic counselling that the authors refer to should be briefly described and discussed in terms of how this may be similar, different to, or overlapping with the genetic counselling provided by individuals formally recognised as genetic counsellors due to completion of extensive training and registration processes.

Thank you for highlighting this important point regarding the role of genetic counseling in nursing. We have revised the manuscript to include a more detailed description of the genetic counseling activities provided by nurses. Specifically, we have clarified how nursing-based genetic counseling aligns with, differs from, and complements the roles performed by formally recognized genetic counselors.

In the revised text, we explain that while nurses engaged in genetic counseling offer essential support in areas such as basic genetic education, initial risk assessment, and emotional support, their role typically remains less specialized than that of certified genetic counselors. Certified genetic counselors undergo extensive training and registration processes, equipping them to provide in-depth psychosocial counseling and interpret complex genetic data. By contrast, nurses integrate genetic counseling as part of broader patient care responsibilities, especially in specialized fields like oncology and cardiology, where they prepare patients for genetic testing, explain results, and guide follow-up care.

This distinction has been integrated into the introduction and relevant sections to contextualize the nursing role in genetic counseling within the broader healthcare framework, emphasizing how nurses contribute to genetic services while respecting the specialized expertise of formally recognized genetic counselors.

Line 64. A rephrasing of this sentence is needed as G2NA has not developed ‘competencies’ in integration and leadership

Thank you for the feedback. We have revised the sentence to accurately reflect G2NA’s role, removing the reference to the development of competencies in integration and leadership. The updated phrasing now highlights G2NA’s focus on supporting genomics integration within nursing practice through international collaboration and leadership.

Line 96 more information is needed on the ‘expert nurse in genetics and genomics’ and how establishing this role will help address education of nursing graduates (as described in the sentence before).

We appreciate your valuable suggestion to elaborate on the role of the 'Expert Nurse in Genetics and Genomics.' In response, we have expanded the introduction to detail how establishing this role would bridge theoretical education and clinical practice, support ongoing education for both new graduates and practicing nurses, and ultimately enhance patient care. This addition clarifies the specific ways in which the expert nurse role would address gaps in nursing education related to genetics and genomics.

Materials and Methods

If the predefined evaluation matrix (line 164) is different to the data extraction information (line 168-175) please provide details of what this matrix contains. If they are the same thing, please be consistent in your terminology.

Thank you for your feedback regarding the terminology in the 'Materials and Methods' section. We have clarified that the 'evaluation matrix' and 'data extraction' are the same tool and have revised the text to reflect consistent terminology. The updated description now specifies that the predefined evaluation matrix was used both to identify relevant courses and to extract detailed data, enhancing clarity and alignment with your suggestion.

Please provide details of the team’s interpretation of ‘basic vs advanced’ content (line 184).

We appreciate your insightful suggestion to elaborate on our interpretation of 'basic vs advanced' genetic and genomic content. We have added a clarification in the 'Data Analysis' section detailing our criteria for categorizing 'basic' and 'advanced' genetic/genomic content to address the reviewer's request for more information on these distinctions.

In 3.1 examples of content (‘specialized areas that are crucial for clinical practice’ line 212) are provided. Did the authors start the study with curricula content that they expected to find (eg based on the nursing/genomics literature or established nursing competencies in genomics eg Eurogentest)? [If they did, this detail should be included in the Methods section. Absence of expected topics in curriculum document(s) should be included in the findings]

 We thank you for this insightful suggestion. To align our study with established nursing competencies in genetics and genòmics we identified an initial set of core tòpics. We have now included this information in the Methods section to clarify our approach. Additionally, we have revised the Findings section to indicate where these expected topics were notably absent, underscoring the educational gaps in nursing curricula regarding genetics and genomics in Spain.

Results

If 118 centres align to 65 universities, how are the 12 that do not teach genetics/genomics distributed? Is it that a small number of universities don’t teach it across any of their affiliates and campuses, or are there up to 12 universities where one site does not include it?

Thank you for highlighting this point. The 12 centers that do not include genetics/genomics content are indeed isolated cases within larger universities, where other affiliates may teach these subjects. This inconsistency indicates that some universities have not fully standardized genetics/genomics education across all affiliated centers. To our knowledge, universities generally have curriculum committees or oversight mechanisms aimed at ensuring consistency; however, the degree of enforcement and monitoring across different sites appears to vary. We have clarified this distribution in the revised manuscript to provide a more comprehensive understanding of the teaching landscape.

Line 273 suggests that it is the latter. Do universities have mechanisms to identify and deal with inconsistencies in teaching the same course across multiple sites?

Thank you for your valuable observation. We have added a paragraph in the discussion section to address the lack of harmonization in genetics and genomics education within Spanish nursing programs, particularly concerning variability across affiliated campuses. This paragraph now also discusses the need to strengthen institutional oversight mechanisms and to adopt national guidelines to improve consistency in training across all campuses within the same university.

Line 404 indicates that some institutions were excluded from the analysis. This detail and number of institutions needs to be included in the results section.

We appreciate your valuable feedback. In response, we have clarified the exclusions within the "Materials and Methods" section under data collection procedure. Specifically, we noted that of the 118 initially identified university centers, 12 were excluded from the analysis due to the absence of genetic or genomic content in their teaching guides. We also confirmed that no other exclusions were necessary, as the remaining data fully met our inclusion criteria. Thank you for your suggestion, which has enhanced the clarity of our methodology.

The inclusion of anonymised examples of how universities describe the content that will be covered in courses would be helpful to illustrate difference between universities.

Thank you for your insightful comment regarding the inclusion of anonymized examples to illustrate how universities describe the content covered in genetics and genomics courses. We agree that such examples would enhance the manuscript by providing concrete illustrations of the variability among institutions. In response to your suggestion, we have revised the manuscript to include a new subsection (Section 3.6) titled "Examples of Course Content Descriptions in Genetics and Genomics." In this section, we present anonymized excerpts from course syllabi of different universities, highlighting the differences in how genetics and genomics are integrated into nursing programs. These examples demonstrate the range from basic introductions within general biology courses to dedicated, in-depth courses on clinical genetics and genetic counseling. We believe that this addition strengthens our analysis and provides readers with a clearer understanding of the disparities in genetics and genomics education across Spanish nursing curricula.

Discussion

Much of the current discussion is a repetition of the findings. A more rigorous discussion of the results is needed and could draw on wider educational literature. For example:

-          line 215-216 ‘although these are usually integrated into broader modules and often do not receive dedicated focus.’ This suggests that integration may be a less effective teaching approach for genetics/genomics. What is the evidence around stand-alone vs integrated genetics/genomics content on student knowledge (or other outcomes?)

Thank you for your insightful comment regarding the effectiveness of integrated versus stand-alone genetics/genomics education. In response, we have expanded Section 4.2 of our manuscript to include a discussion of educational literature addressing this issue. We have cited several studies that provide evidence on how stand-alone courses in genetics and genomics lead to better student outcomes, such as higher competency levels, improved knowledge retention, and increased confidence in applying genomic knowledge. Our findings align with this evidence, suggesting that while integrated curricula offer contextual learning, they may not provide the necessary depth for complex subjects like genetics and genomics.

-          Line 221 calculation of genetics/genomics ECTS and the statement that currently the figure is insufficient, provides an opportunity to discuss whether there should be a recommended number of ECTS and what this number should be. Will quantity (of teaching) be a benchmark for quality?

Thank you for your valuable feedback regarding the ECTS credits allocation for genetics and genomics. We have expanded Section 4.3 to address your suggestions. Specifically:

  • Recommended ECTS Allocation: Recognizing the current allocation of 3–6 ECTS credits is insufficient, we propose that nursing programs allocate at least 10–12 ECTS credits (approximately 4–5% of total degree credits) to genetics and genomics. This recommendation is based on the complexity and significance of these subjects in modern healthcare.
  • Quantity vs. Quality: We acknowledge that increasing credits alone may not guarantee improved educational outcomes. Therefore, we discuss how educational quality also depends on effective teaching methodologies, faculty expertise, curriculum integration, and practical experiences.

We believe these additions address your concerns and enhance the manuscript.

-          Line 231-233 “without practical activities that allow students to develop applied competencies. This approach negatively impacts the development of practical skills that are essential for applying this knowledge in direct patient care.” Without evidence that skill development or application of knowledge is impacted at these Spanish universities, this sentence in the findings section should be written as an assumption rather than a statement of fact. The authors have an opportunity in the discussion to reference other approaches to teaching genetics/genomics that have been shown to be effective.

Thank you for your insightful feedback regarding the statement in lines 231-233. You are correct that our study did not directly measure the impact of the lack of practical activities on skill development at the Spanish universities analyzed. In response, we have revised the sentence in the findings section to reflect this as an inference:

  • Revised Sentence: "This approach may hinder the development of practical skills that are essential for applying this knowledge in direct patient care."

Additionally, we have expanded the discussion section to reference other teaching approaches in genetics/genomics education that have been shown to be effective. We have included studies that highlight the importance of practical activities, such as simulations and case studies, in enhancing skill development and the application of knowledge in clinical settings.

We appreciate your suggestion, which has helped us improve the clarity and accuracy of our manuscript.

-          Section 3.4 standardisation. Are there other content areas of the Spanish nursing curricula where there is standardisation and if there are, how has this been achieved?

The reviewer is assuming that at least some of the curriculum documents would contain information on assessment. It would also be helpful to know if any made explicit reference to assessment of genetic/genomic knowledge and/or skills as assessment is often discussed as a driver for learning.

Thank you for your insightful comments regarding Section 3.4 on standardization. In response, we have made the following concise revisions:

  • Section 3.4:
    • Added that other subjects like Anatomy, Physiology, and Pharmacology are standardized through national regulations and collaborative efforts, showing that standardization in genetics and genomics is achievable. Noted that curriculum documents seldom include explicit assessment of genetic/genomic knowledge and skills. Recognizing that assessment drives learning, we emphasize the need for targeted assessment strategies.
  • Section 4.5:
    • Suggested that applying standardization strategies from other subjects—such as developing national competencies and enforcing accreditation requirements—could harmonize genetics and genomics education across institutions. Highlighted the crucial role of explicit assessment methods in reinforcing the importance of genetics and genomics and enhancing student engagement.

These additions address your concerns by illustrating how standardization has been achieved in other curriculum areas and underscoring the role of assessment in genetics/genomics education.

The authors make a number of recommendations (subsection 4.8) and call for the establishment of national minimum standards for genetics and genomics education (Conclusions line 431). The authors should propose a mechanism(s) as to how these recommendations might be achieved and outline, for example, the individuals, organisations and/or professional bodies that need to be involved to drive strategy and change policy. Given that many other countries are likely to be in a similar situation, readers may be interested to understand how these challenges could be tackled.

We appreciate your comments concerning our recommendations and the need to outline mechanisms for their implementation. In response, we have expanded Section 4.9 by integrating specific strategies directly into each recommendation, creating a continuous and concise narrative. This approach details how our proposals can be actualized and identifies the key stakeholders involved in driving strategy and policy changes.

By incorporating these mechanisms, we aim to provide a clear and actionable plan that not only addresses the challenges within Spain but also offers a framework applicable to other countries facing similar issues. We believe these additions enhance the practical value of our recommendations and offer readers valuable insights into tackling these challenges.

line 386 If more than one competency framework is required for Spanish nursing education, please provide more information. If this is a typographical error the ‘s’ should be removed from Frameworks

Thank you for your attentive reading of our manuscript. You are correct that in line 386, the term "competency frameworks" was a typographical error. We intended to refer to a single "competency framework." We have corrected the manuscript by removing the 's' to make it singular. We appreciate you bringing this to our attention and have ensured the correction is reflected in the revised version.

Thank you again for your insightful comments, which have significantly enhanced the quality of our paper.

Sincerely,

Round 2

Reviewer 3 Report

Comments and Suggestions for Authors

Thank you for such a comprehensive response to the original review. The inclusion of curriculum examples is a helpful illustration of the variation that you found. I do believe that the revisions have really strengthened the manuscript, and I am sure your work will be of interest to many.

I haven’t reviewed all of the changes to the references; you have indicated that you have checked links etc. However, I have identified one of the new paragraphs where there may be an error. I was interested in the standardisation work in anatomy, physiology and pharmacology.

 “The standardization in these other areas results from collaborative efforts among educational institutions, regulatory bodies, and professional associations, involving regular curriculum reviews and adherence to national educational frameworks [24,25].

I was expecting references that describe these activities in Spain. The use of refs 24 and 25 [line 329] appears incorrect as these are paper from the United States. [References 24 and 25 are used correctly elsewhere in the paper].

Author Response

Dear Reviewer,

Thank you for your insightful and constructive feedback on the revised manuscript. I am delighted that the revisions have strengthened the manuscript and that you found the examples illustrating curriculum variability helpful.

I deeply appreciate your observation regarding the references in the section discussing standardization efforts in anatomy, physiology, and pharmacology. After carefully reviewing all references in the manuscript, I noticed the discrepancy you pointed out. I have now replaced the references in question with sources that accurately reflect standardization efforts in Spain. Additionally, I have meticulously reviewed all citations to ensure their accuracy and correct placement throughout the manuscript.

Thank you again for your valuable comments and your attention to detail, which have helped to further improve the rigor and clarity of the manuscript. I remain grateful for your time and expertise in evaluating this work, and I am confident that the updated version is now better aligned with the intended focus.

Kind regards,